# Localization of GNSS Spoofing Interference Source Based on a Moving Array Antenna

**Rui Liu** [1,2,*], **Zhiwei Yang** [1], **Qidong Chen** [1,2], **Guisheng Liao** [1] and **Qinglin Zhu** [2]

1   National Laboratory of Radar Signal Processing, Xidian University, Xi'an 710000, China; yangzw@xidian.edu.cn (Z.Y.); chenqd@crirp.ac.cn (Q.C.); liaogs@xidian.edu.cn (G.L.)
2   China Research Institute of Radiowave Propagation, Qingdao 266000, China; zhuql@crirp.ac.cn
*   Correspondence: liur@crirp.ac.cn

**Abstract:** GNSS spoofing interference utilizes falsified navigation signals to launch attacks on GNSS systems, posing a significant threat to applications that rely on GNSS signals for positioning, navigation, and time services. Therefore, achieving effective localization of the sources causing spoofing interference is crucial in ensuring the secure operation of GNSS. This article proposes a method for locating GNSS spoofing interference sources using a moving array antenna. Firstly, the proposed method utilizes the inherent characteristics of the double-differenced carrier phase from the deception signals received by the array antenna to effectively extract the spoofing signals. Subsequently, by moving the antenna array, the original carrier phase single-difference data of multiple observation points for deception signals are fused to provide a cost function for direct localization of spoofing interference, and a solution method for the cost function is designed. The proposed method addresses the challenge of extracting and localizing GNSS spoofing interference weak signals, effectively avoiding the data correlation of traditional two-step methods for DOA estimation parameters and ensuring the location accuracy of spoofing interference and the robustness of the method. The effectiveness of the proposed method has been validated through simulation experiments, and its adaptability to factors such as errors in carrier phase measurements has been examined. The method exhibits strong applicability and is well-suited for the hardware platform of the GNSS nulling antenna, thereby enabling it to possess simultaneous capabilities in both anti-interference and spoofing interference localization.

**Keywords:** GNSS spoofing interference; spoofing interference localization; array antenna; double-differenced carrier phase

## 1. Introduction

The term global navigation satellite system (GNSS) typically refers to various global satellite navigation systems and their enhanced systems, including GPS in the United States, Galileo in Europe, GLONASS in Russia, and Beidou in China [1]. Due to the weak GNSS received power of $-130$ dBm, the GNSS spectrum is heavily immersed in environmental noise and exhibits elevated susceptibility to electromagnetic interference [2,3]. Jamming and spoofing are the two most significant forms of intentional interference, with spoofing involving the generation of counterfeit satellite navigation signals or the retransmission of authentic satellite navigation signals [4,5]. Spoofing can manipulate the code phase, carrier phase, message content, and additional parameters depending on the spoofing intention. This allows the receiver to unknowingly capture and track the spoofing signals, leading to erroneous positioning results and enabling control over the receiver [6]. Compared to jamming, spoofing exhibits a power level akin to authentic satellite signals but with lower signal strength, better concealment, and greater potential for harm [7]. The aforementioned characteristics of spoofing interference pose significant challenges for wireless detection and localization approaches, and hence, this paper focuses on the problem of localizing GNSS spoofing sources.

### 1.1. Related Work

Currently, numerous anti-spoofing technologies [8] are available for the detection, identification, and suppression of spoofing signals in GNSS systems. These include in-band power monitoring [9], signal quality monitoring [10], receiver timing monitoring [11], navigation message authentication [12], multi-antenna technology [13], and multi-receiver technology [14], among others. However, there have been limited studies on the localization of GNSS spoofers.

The commonly employed methods for localizing GNSS interference sources include direction of arrival (DOA) methods [15,16], time difference of arrival (TDOA) methods [17,18], and received signal strength (RSS) methods [19,20]. The signal strength of spoofing interference is equivalent to the real GNSS signal strength, and the signal-to-noise ratio (SNR) of the signal can be as low as $-19$ dB, which makes it difficult to extract the DOA, TDOA, and RSS of the signal. DOA methods frequently use algorithms such as CAPON and MUSIC to estimate signal arrival angles, which cannot efficiently extract the DOA of spoofing interference signals in the aforementioned low-SNR conditions. The extraction of TDOA with weak spoofing interference requires long-term data correlation operations, which requires TDOA methods to collect and transmit a large amount of raw data. It is difficult to extract the RSS because the spoofing signal is submerged in noise. Importantly, due to the extreme similarity between the spoofing signal and the real signal, it is difficult to distinguish the signal features of the real and spoofing signals from the extracted ones, even if the above localization methods implement signal feature estimation. For example, the navigation receiver can compute the C/N0 of the spoofing signal to achieve the RSS measurement, but it cannot distinguish the C/N0 of the real satellite signal from that of the deception signal, and the DOA and TDOA methods face similar problems.

The deception localization methods proposed by [21,22] have shown relative effectiveness; however, their establishment relies on stringent restrictive assumptions, thereby impacting the practicality of these methods. Reference [21] is a low-cost deception interference localization method, but the establishment of the method requires accumulating observation data under the assumption that the deception interference source is stationary and the replay delay does not modify over time or has the same shift, which is a relatively strict assumption condition. In addition, reference [21] can only realize the repeater spoofing localization, while the generative deception source is more common. Reference [22] is a deception interference localization method based on TDOA, and the article utilizes the geometry and known positions of multiple static GPS receivers distributed within the power substation. The establishment of this method requires the construction of a GNSS monitoring sensor network and requires the receivers in the network to return relevant peak information, which brings certain limitations to the practical application of the method.

This article adopts a direct localization method, while the traditional localization method adopts a two-step processing [23]. The two-step method first extracts measurement parameters (such as DOA and TDOA) and then estimates the source location. Direct localization achieves source localization without estimating signal parameters. This type of method can avoid correlation problems [24], and its localization accuracy has been proven to be significantly higher than traditional two-step methods, especially under low signal-to-noise ratio conditions [25]. In the application of this paper, due to the weak signal of the spoofing signal and the dynamic monitoring scenario, the direct localization method is used to localize the GNSS spoofing interference, which improves the localization accuracy and robustness of the method in this scenario.

The article proposes a moving array antenna-based method aiming to address the issue of locating GNSS spoofing interference sources. This method can be implemented on the widely used hardware platforms of GNSS nulling antenna [26], enabling it to possess simultaneous capabilities of anti-interference and deception interference detection and localization.

### 1.2. Our Contributions

The article presents a method to implement the recommended antenna model and navigation signal capture tracking method and proposes our approach based on this. The

proposed method leverages the consistent emission of deceptive satellite signals from a common source of interference, effectively extracting and confirming these misleading signals through statistical analysis of the double-differenced carrier phase. The article then formulates a cost function to address the direct positioning of spoofing interference with a moving antenna array and integrating carrier phase single-difference raw observation data from multiple observation points, while also presenting a solution methodology for this cost function. Additionally, the article also provides the Cramer Rao Bound (CRB) for the localization method of GNSS spoofing interference sources.

The effectiveness of the proposed method has been verified through simulations, and the simulation results demonstrate that the proposed method can effectively achieve the detection, extraction, and localization of GNSS spoofing interference sources. In comparison to the conventional two-step methods, the proposed method has stronger adaptability toward larger carrier phase measurement errors (weaker deceptive interference signal strength) and antenna positioning errors. Moreover, the localization accuracy achieved by this approach closely approximates that of the global search method and CRB. It also examines the performance of the proposed method on typical localization observation paths and suggests optimal localization paths. By conducting experiments to assess the effect of varying the number of observation points on the localization accuracy, we propose optimal configurations for the number of observation points in this particular scenario.

The subsequent sections of this paper are structured as follows. Section 2 initially presents the system model of our approach, encompassing the antenna model and methods for navigation signal acquisition and tracking. Building upon this foundation, Section 3 proposes a method for localizing GNSS spoofing interference sources using a moving array antenna. Section 4 of this paper presents the experimental chapter for the proposed method, which validates the effectiveness of the method and performs a comprehensive and systematic simulation analysis of the key factors affecting its performance. Section 5 is devoted to the conclusion of this paper.

## 2. System Model and Signal Processing Methods

This article presents a method for localizing GNSS spoofing interference sources using a moving array antenna, which is commonly employed in the scenario depicted in Figure 1. In this scenario, the unmanned aerial vehicle (UAV) for spoofing interference monitoring is equipped with an array antenna and a receiver dedicated to monitoring spoofing interference. It employs mobile monitoring to collect authentic satellite signals as well as navigation deceptive spoofing signals, which are then processed using the methodology proposed in this paper to effectively monitor and locate instances of GNSS deceptive interference behavior. The proposed method is also applicable to the monitoring and localization of GNSS spoofing interference sources in vehicular scenarios.

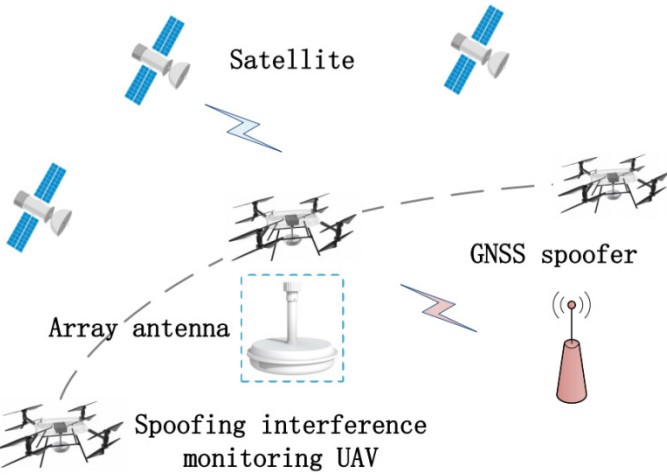

**Figure 1.** Typical application scenarios for the method.

### 2.1. Antenna Model

The proposed method is applicable to any array antenna operating in the navigation frequency band, and it is recommended to utilize the 6 + 1 array antenna model described in reference [27]. This particular model has been widely adopted to design GNSS anti-jamming nulling antennas, as illustrated in Figure 2. The design effectively achieves a trade-off between signal measurement accuracy and antenna availability by integrating relatively more elements within a compact space. The model consists of 7 antenna elements, where element 0 serves as the reference and is located at the center of the antenna structure, while elements 1–6 are uniformly distributed along a circular path of a radius, denoted by $R$. To ensure that there is no spatial ambiguity between adjacent array elements, the baseline length between elements is equal to half of the carrier wavelength.

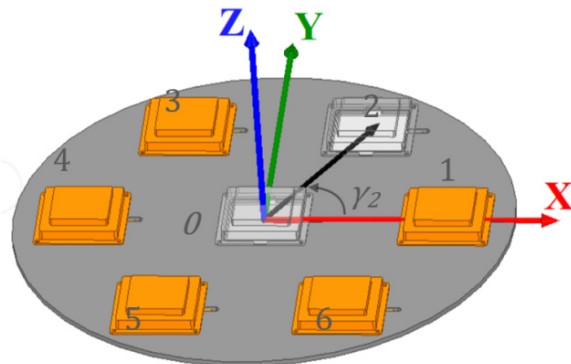

**Figure 2.** The 6 + 1 antenna model.

As depicted in Figure 2, we build a Cartesian coordinate system with the position of antenna element 0 as the origin. The $x$-axis is oriented positively from element 0 towards element 1, while the $y$-axis is oriented positively from element 0 towards the midpoint between elements 2 and 3. Moreover, the $z$-axis is perpendicular to the plane of the antenna model. The angle $\gamma_i = 2\pi(i-1)/6$, $i = 1, 2, \cdots, 6$ represents the inclination of element $i$ with respect to the positive $x$-axis, thus allowing us to express the position vector of the array elements as $\boldsymbol{p_i} = [Rcos\gamma_i, Rsin\gamma_i, 0]^T$. This diagram provides a schematic depiction of $i = 2$ positioned at an angle $\gamma_2$.

### 2.2. Signal Acquisition and Tracking Methods

After implementing the antenna model described in Section 2.1 for signal reception, this method devises a signal receiving and processing process, as depicted in Figure 3, to effectively handle all received genuine and spoofing GNSS signals at each antenna element, thereby the acquisition, tracking, and carrier phase measurement of all the signals can be accurately achieved. Each array element of the antenna array depicted in Figure 3 corresponds to a single radio frequency (RF) channel, enabling RF signal conditioning, down-conversion, intermediate frequency (IF) signal filtering and amplification, and analog-to-digital (AD) conversion to obtain the digital IF signal corresponding to each element.

As shown in Figure 3, we have designed an efficient satellite signal acquisition method suitable for the scenario described in this paper. This method performs signal acquisition only on the signals received from array element 0 to obtain observation vectors $\left[\tau_0^1, \cdots \tau_0^h, \cdots, \tau_0^K\right]$ and $\left[f_0^1, \cdots f_0^h, \cdots, f_0^K\right]$, where $\tau_0^h$ and $f_0^h$ represent a coarse estimate of the code phase and carrier frequency measurement for satellite $h$, respectively. Given that the inter-element spacing is within half of the carrier wavelength, this approach uses the acquisition observation vector of element 0 directly as input to track the signals from the other antenna elements, thereby circumventing the need for signal acquisition processes involving the remaining 6 elements. At the same time, the algorithm in reference [28] can be used to further reduce the implementation cost of the method. This algorithm is an

improved parallel code phase acquisition (PCA) algorithm that can speed up the acquisition process while slightly reducing the detection probability.

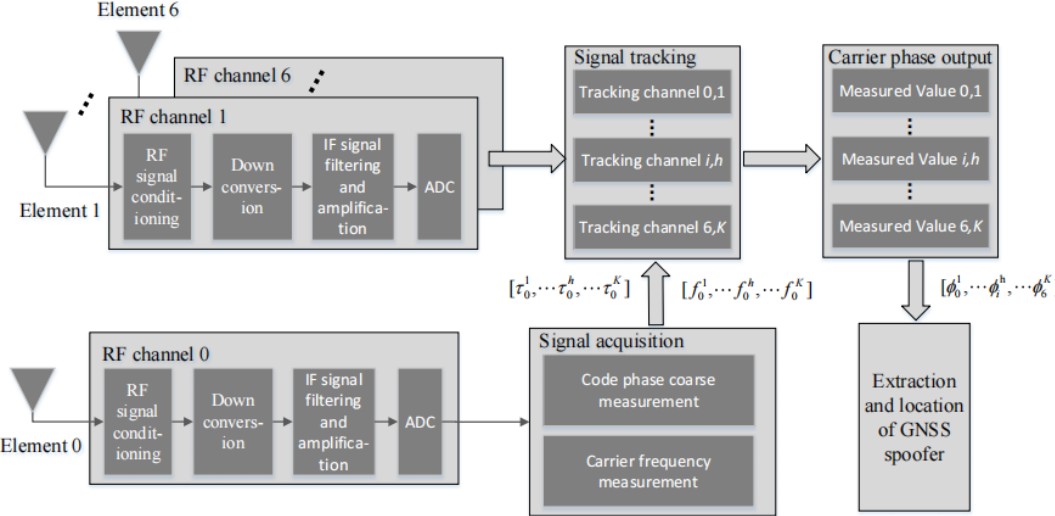

**Figure 3.** Schematic diagram of signal reception and processing.

The signal tracking process in Figure 3 involves assigning a tracking channel to each satellite signal received by each antenna element, where the tracking channel *i, h* is the channel to satellite *h* received by element *i*. The tracking process enables the measurement of the carrier phase and generates the observed vector $\left[\phi_0^1, \cdots \phi_i^h, \cdots, \phi_6^K\right]$ of carrier phase, where $\phi_i^h$ represents the measured value of satellite *h*'s carrier phase received by array element *i*. The method described in this paper leverages the aforementioned carrier phase observations for efficient extraction and localization of GNSS spoofing interference.

Figure 4 depicts the signal reception loop design employed for the tracking channel. In Figure 4, the loop consists of a code loop located at its upper part and a carrier loop located at its lower part. The signal tracking method employed in this paper follows the classical approach described in reference [29], which is widely adopted for GNSS receiver design.

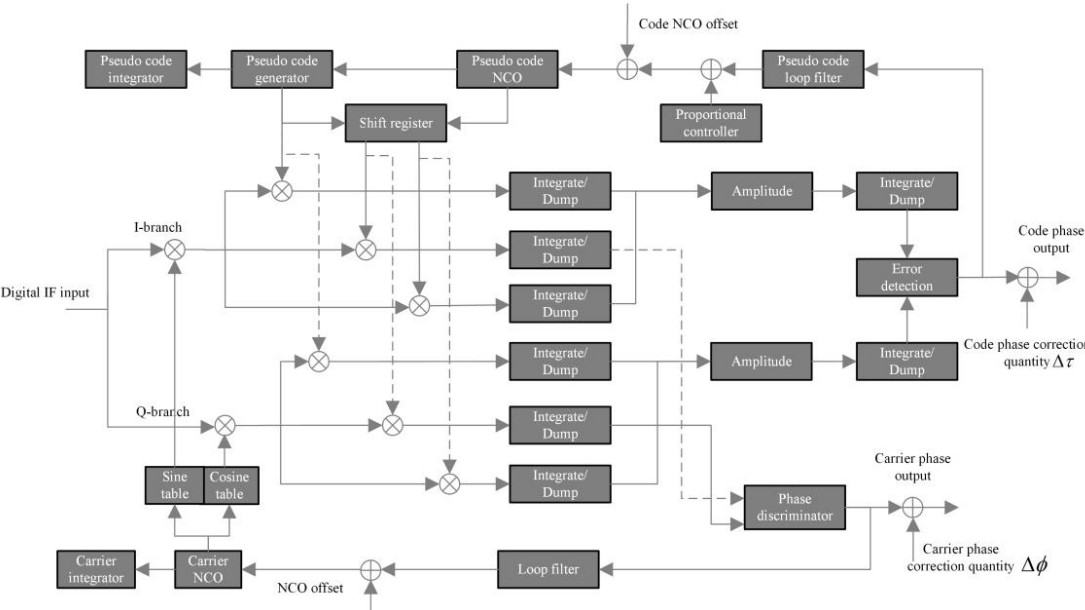

**Figure 4.** Structural diagram of the signal tracking loop.

The correction parameters for the code phase and carrier phase, denoted as $\Delta\tau$ and $\Delta\phi$, respectively, in Figure 4, are calculated according to the method detailed in reference [30]. The proposed method, which can be implemented on a GNSS nulling antenna hardware platform, enables simultaneous spoofing interference detection and localization with anti-jamming. However, adaptive beamforming modifies the array pattern in real time, which has the potential to introduce phase–center bias into the antenna array. Since we need to maintain the accurate measurement of the carrier phase in the application scenario of this paper, these deviations must be compensated for to reduce the carrier phase measurement error of the GNSS receiver. Reference [30] gives the calculation method of the code phase and carrier phase correction amount to compensate for the above error. In this paper, we use the carrier phase correction amount to correct the carrier phase measurements of the tracking loop and obtain accurate carrier phase measurements.

## 3. Proposed Methods

The power of the spoofing interference is similar to that of the genuine satellite signal compared to the suppressed interference. Due to the fact that deceptive interference is submerged in noise, angle estimation methods based on antenna arrays are difficult to achieve efficient direction finding for deceptive interference. The article utilizes the spread-spectrum characteristics of deceptive interference and implements conditioning, acquisition, and tracking of all GNSS signals received by the antenna array using the system and signal processing model described in Section 2, thereby obtaining measurements of the carrier phase. The paper proposes a moving array antenna-based method for localizing GNSS spoofing interference sources. The method first uses double-differenced carrier phase observations to distinguish between genuine GNSS signals and spoofing signals, thereby confirming and extracting the spurious spoofing satellite signals. Subsequently, this approach combines the raw carrier phase single-difference measurements from multiple observation points of an array antenna to achieve direct localization of the spoofing source. A comprehensive description of this methodology will be provided in this section.

### 3.1. Extraction of Spoofing Interference Based on the Double-Differenced Carrier Phase

After successfully receiving both authentic and spoofing GNSS signals, this method effectively extracts and confirms the deceptive signals among the previously received signals, thereby enabling subsequent localization of the spoofing source. The proposed method leverages the inherent characteristics of same-origin signals and the spread spectrum properties of spoofing interference to effectively extract the deceptive interference by observing and statistically analyzing the double-differenced carrier phase feature. According to the content referenced in [31], we present the observation model for the carrier phase measurement value obtained in Section 2 as follows:

$$\phi = \frac{2\pi}{\lambda}(\rho + \delta\rho + c(\delta tr - \delta ts) + \delta\rho c + T - I + \delta\rho m) - 2\pi N + \varepsilon \tag{1}$$

in which $\phi$ is the carrier phase observation value of the navigation signal, $\lambda$ is the carrier wavelength of the signal, $\rho$ is the distance from the satellite to the receiver antenna, $\delta\rho$ is the orbit error of the satellite, $c$ is the speed of light, $\delta tr$ is the receiver clock difference, $\delta ts$ is the satellite clock difference, $\delta\rho c$ is the error caused by the receiver cable, $T$ is the error caused by the troposphere, $I$ is the error caused by the ionosphere, $\delta\rho m$ is the error caused by the multi-path effect, $N$ is the number of complete cycles of carrier phase, and $\varepsilon$ is the measurement noise. The units of $\phi$ and $\varepsilon$ are rad, the units of $\rho$, $\delta\rho$, $\delta\rho c$, $T$, $I$, and $\delta\rho m$ are m, the units of $\delta tr$ and $\delta ts$ are s, the unit of $c$ is m/s, and the unit of $N$ is 1. According to Equation (1), the carrier phase single-difference between array element $i$ and array element 0 can be calculated, i.e., $\varphi_i = \phi_i - \phi_0$. Taking satellite signal $h$ as an example, its single-difference calculation equation is as follows.

$$\varphi_i^h = \phi_i^h - \phi_0^h = \frac{2\pi}{\lambda}\left(\rho_i^h + \delta\rho_i^h + c(\delta tr_i^h - \delta ts_i^h) + \delta\rho c_i^h + T_i^h - I_i^h + \delta\rho m_i^h\right) - 2\pi N_i^h + \\ \varepsilon_i^h - \frac{2\pi}{\lambda}\left(\rho_0^h + \delta\rho_0^h + c(\delta tr_0^h - \delta ts_0^h) + \delta\rho c_0^h + T_0^h - I_0^h + \delta\rho m_0^h\right) + 2\pi N_0^h - \varepsilon_0^h \tag{2}$$

According to reference [32], for the same signal source, due to the very close distance between the array elements, it can be assumed that the parameters $\delta\rho$, $T$, $I$, and $\delta ts$ between the array elements are the same. By implementing an equal-length design for the antenna element feed lines, an equal-length layout design for the sampling circuit, and digital delay correction for each channel, it becomes feasible to maintain consistent $\delta\rho c$ values across all array elements. By employing the same highly stable crystal oscillator as the time-frequency reference for the sampling circuit, the RF circuit, and the navigation signal processing circuit associated with each array element, the parameter $\delta tr$ between each array element can be consistent. Moreover, since the distances between antenna array elements are designed without integer ambiguities, the formulation of the carrier phase single-difference measurement $\varphi_i^h$ can be further simplified as follows.

$$\varphi_i^h = \frac{2\pi}{\lambda}(\rho_i^h - \rho_0^h) + \varepsilon_i^h - \varepsilon_0^h \tag{3}$$

The conditions described here are overly idealized, but they can be achieved in practice. Reference [32] adopts a similar approach to this article to achieve the above conditions. The software-defined radio (SDR) platform proposed in reference [26] implements the above equal-length conditions, which can be used for the implementation and testing of the method in this paper. In practical applications, the GNSS signal source can be used to calibrate multiple hardware channels (including antenna feeders), and the calibration diagram is shown in Figure 5.

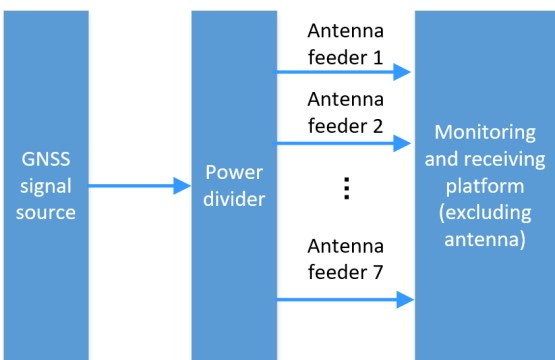

**Figure 5.** Calibration diagram of multiple hardware channels.

In this method, the seven channels corresponding to the 6 + 1 array antenna track the same navigation satellite signal output by the GNSS signal source and output the carrier phase measurement differences between the hardware channels of element 1 to element 6 and the hardware channel of element 0 as the correction values. In practice, these correction values are used to achieve a consistent correction for each channel.

In this paper, the GNSS spoofing interference extraction is performed using double-difference carrier phase measurements. For satellite signals $k$ and $h$, the carrier phase double-difference equation for array element i is defined as follows.

$$\Delta\varphi_i^{k,h} = \varphi_i^k - \varphi_i^h = \frac{2\pi}{\lambda}\left(\rho_i^k - \rho_0^k\right) + \varepsilon_i^k - \varepsilon_0^k - \frac{2\pi}{\lambda}\left(\rho_i^h - \rho_0^h\right) - \varepsilon_i^h + \varepsilon_0^h \tag{4}$$

If the signals from satellites $k$ and $h$ originate from the same spoofing source, then the propagation paths of both satellite signals from transmitter to receiver are identical. Therefore, the carrier phase double-difference $\Delta\varphi_i^{kh}$ can be simplified as follows.

$$\Delta\varphi_i^{k,h} = (\varepsilon_i^k - \varepsilon_0^k) - (\varepsilon_i^h - \varepsilon_0^h) \tag{5}$$

The tracking error of the carrier phase is typically less than 1% of the carrier wavelength [33], that is, $|\varepsilon| \leq 0.005 \times 2\pi$ rad, thus we set the maximum value of $8|\varepsilon|$ as the threshold for identifying and extracting spoofing interference, that is, $Th = 0.04 \times 2\pi$ rad. If the double-differenced carrier phase observation between the satellite signal $k$ and the satellite signal $h$ is less than the threshold value $Th$, then it is possible that both the satellite signals originate from the same spoofing interference. In practical application scenarios, considering the method's accuracy and real-time nature, we establish that signals from two satellites originate from the same spoofing interference source when there are continuous five-carrier phase double-difference observation values below a certain threshold.

Considering that the carrier phase may fluctuate in dynamic scenarios, we use the statistical method for carrier phase measurement errors in reference [34]. When implementing the proposed method in this paper, we can set up two hardware channels for carrier phase measurement on the same antenna element and can monitor and compile statistics on the carrier phase measurement error $\varepsilon$, in real-time. Then, $8|\varepsilon|$, corresponding to the measured $\varepsilon$, is used as the interference extraction threshold to improve the effectiveness of the spoofing interference extraction and confirmation in this paper.

### 3.2. Spoofing Interference Direct Location Based on Carrier Phase Single-Difference

The method described in Section 3.1 enables the extraction and determination of spoofing interference signals from the received GNSS signals by the array antenna. Expanding on this, in this section, we directly obtain the position of the spoofing interference source by fusing raw carrier phase observations using a moving antenna array. Due to the dynamic observing scenario in this paper, there are some carrier phase observation errors and array antenna positioning errors. We use carrier phase single-difference measurements to construct the observation equations for the spoofing source and directly determine the location of the spoofing source by moving the antenna array and combining the observation equations. Compared with traditional two-stage positioning methods, the direct location method avoids spatial correlation of signal estimation parameters and enhances the robustness of spoofing localization in dynamic scenarios [23].

We set up the scenario model for the spoofing interference source location in this paper, illustrated in Figure 6, employing the same Cartesian coordinate system approach as depicted in Figure 2. In this model, we designate the initial position $p_1$ of antenna element 0 as the origin. $p_t$ represents the vector of coordinates corresponding to the location of the spoofing source. $p_j$ denotes the $j$-th signal acquisition position of element 0, while $M$ signifies the total number of observations.

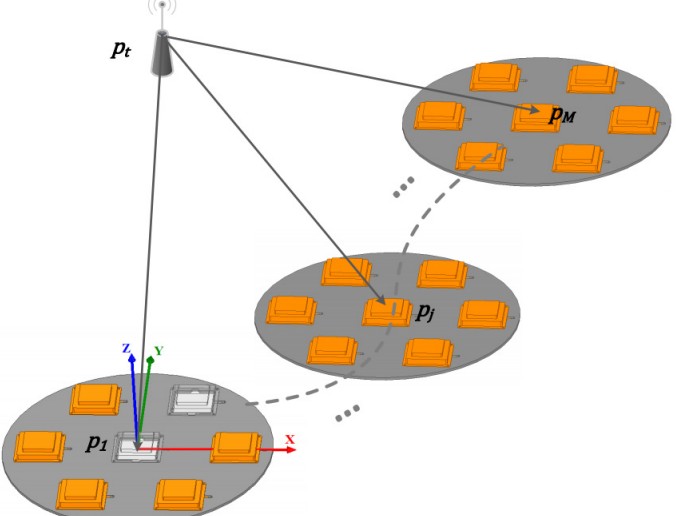

**Figure 6.** Scenario model of spoofing interference direct localization.

The observation equation can be established at observation position $p_j$ by utilizing the carrier phase single-difference measurement $\varphi_{j,i}$ of array element $i$. The formulated expression is as follows:

$$\left\| p_t - p_{j,i} \right\| - \left\| p_t - p_j \right\| = \frac{\lambda \varphi_{j,i}}{2\pi} \tag{6}$$

in which $p_{j,i}$ is the coordinate of array element $i$ at position $p_j = \left[ x_j, y_j, z_j \right]^T$, and $p_{j,i} = p_j + \left[ R\cos\gamma_i, R\sin\gamma_i, 0 \right]^T$. The coordinate $p_t$, which represents the interference source in Equation (6), is the unknown variable to be determined. By substituting $p_t = [x, y, z]^T$, Equation (6) can be reformulated as follows.

$$\sqrt{\left( x - x_{j,i} \right)^2 + \left( y - y_{j,i} \right)^2 + \left( z - z_{j,i} \right)^2} - \sqrt{\left( x - x_j \right)^2 + \left( y - y_j \right)^2 + \left( z - z_j \right)^2} = \frac{\lambda \varphi_{j,i}}{2\pi} \tag{7}$$

We take the equations of position $p_j$ as an example, and we can obtain six nonlinear equations at the position for array elements 1 to 6. In Figure 6, there are $M$ observation positions in the scene, each of which gives the above six nonlinear equations; thus, the number of equations we can obtain is $6M$, where the number of unknowns in the equations is 3 ($x$, $y$, and $z$), and the rest variables are known observations.

We express the nonlinear Equation (7) as follows:

$$g(x, y, z, v_{j,i}) = u_{j,i} \tag{8}$$

In the equation, $g(\cdot)$ represents the nonlinear relationship in Equation (7), while $v_{j,i}$ and $u_{j,i}$, respectively, denote known variables associated with the position of the receiving antenna and carrier phase observation values, where $v_{j,i} = (p_{ji}{}^T, p_j{}^T)^T$, $u_{j,i} = \lambda \varphi_{j,i}/2\pi$. By combining Equation (8) from various positions and array elements, the following system of equations is formulated.

$$\begin{cases} g(x, y, z, v_{1,1}) = u_{1,1} \\ g(x, y, z, v_{1,2}) = u_{1,2} \\ \quad\quad \vdots \\ g(x, y, z, v_{j,i}) = u_{j,i} \\ \quad\quad \vdots \\ g(x, y, z, v_{M,6}) = u_{M,6} \end{cases} \tag{9}$$

We linearize Equation (8) by Taylor expansion and obtain the following result:

$$\begin{aligned} u_{j,i} \approx g(x_{k-1}, y_{k-1}, z_{k-1}, v_{j,i}) + \frac{\partial g(x_{k-1}, y_{k-1}, z_{k-1}, v_{j,i})}{\partial x}(x - x_{k-1}) + \\ \frac{\partial g(x_{k-1}, y_{k-1}, z_{k-1}, v_{j,i})}{\partial y}(y - y_{k-1}) + \frac{\partial g(x_{k-1}, y_{k-1}, z_{k-1}, v_{j,i})}{\partial z}(z - z_{k-1}) \end{aligned} \tag{10}$$

where $\partial g(x_{k-1}, y_{k-1}, z_{k-1}, v_{j,i})/\partial x$ is calculated as follows.

$$\frac{\partial g(x_{k-1}, y_{k-1}, z_{k-1}, v_{j,i})}{\partial x} = \left. \frac{\partial g(x, y, z, v_{j,i})}{\partial x} \right|_{\substack{x = x_{k-1} \\ y = y_{k-1} \\ z = z_{k-1}}} = \left. \frac{\partial g(x, y_{k-1}, z_{k-1}, v_{j,i})}{\partial x} \right|_{x = x_{k-1}} \tag{11}$$

Therefore, the iterative calculation formula for equation system (9) can be provided as follows:

$$H\delta p = b \tag{12}$$

in which

$$H = \begin{bmatrix} \frac{\partial g(x_{k-1}, y_{k-1}, z_{k-1}, v_{1,1})}{\partial x} & \frac{\partial g(x_{k-1}, y_{k-1}, z_{k-1}, v_{1,1})}{\partial y} & \frac{\partial g(x_{k-1}, y_{k-1}, z_{k-1}, v_{1,1})}{\partial z} \\ \vdots & \vdots & \vdots \\ \frac{\partial g(x_{k-1}, y_{k-1}, z_{k-1}, v_{j,i})}{\partial x} & \frac{\partial g(x_{k-1}, y_{k-1}, z_{k-1}, v_{j,i})}{\partial y} & \frac{\partial g(x_{k-1}, y_{k-1}, z_{k-1}, v_{j,i})}{\partial z} \\ \vdots & \vdots & \vdots \\ \frac{\partial g(x_{k-1}, y_{k-1}, z_{k-1}, v_{M,6})}{\partial x} & \frac{\partial g(x_{k-1}, y_{k-1}, z_{k-1}, v_{M,6})}{\partial y} & \frac{\partial g(x_{k-1}, y_{k-1}, z_{k-1}, v_{M,6})}{\partial z} \end{bmatrix} \tag{13}$$

$$\delta p = p_t - p_{t,k-1} = \begin{bmatrix} x \\ y \\ z \end{bmatrix} - \begin{bmatrix} x_{k-1} \\ y_{k-1} \\ z_{k-1} \end{bmatrix} \tag{14}$$

$$b = \begin{bmatrix} u_{1,1} - g(x_{k-1}, y_{k-1}, z_{k-1}, v_{11}) \\ \vdots \\ u_{j,i} - g(x_{k-1}, y_{k-1}, z_{k-1}, v_{ji}) \\ \vdots \\ u_{M,6} - g(x_{k-1}, y_{k-1}, z_{k-1}, v_{M6}) \end{bmatrix} \tag{15}$$

Among them, $p_{t,k-1}$ represents the spoofer's location after $k-1$ iterations, while $H$ denotes the Jacobian matrix. In order to achieve an iterative solution, solve $\delta p$ in Equation (12). By adopting a least-squares solution approach, we formulate the cost function as follows.

$$F(\delta p) \equiv \|H\delta p - b\|^2 = (H\delta p - b)^T (H\delta p - b) \tag{16}$$

By expanding the cost function, we obtain Equation (17).

$$\begin{aligned} F(\delta p) &= \delta p^T H^T H \delta p - \delta p^T H^T b - b^T H \delta p + b^T b \\ &= \delta p^T H^T H \delta p - 2 \delta p^T H^T b + b^T b \end{aligned} \tag{17}$$

Due to the symmetric positive definiteness and invertibility of the matrix $H^T H$, the minimum of $F(\delta p)$ is guaranteed. By computing the derivative of $F(\delta p)$ with respect to $\delta p$, we can derive the following equation.

$$\frac{dF(\delta p)}{d(\delta p)} = 2H^T H \delta p - H^T b \tag{18}$$

The value of $\delta p$, when the above derivative value equals 0, is the desired iterative solution, which is formulated as follows.

$$\delta p = \left(H^T H\right)^{-1} H^T b \tag{19}$$

Upon obtaining $\delta p$, the solution of Equation (9) can be updated from $p_{t,k-1}$ to $p_{t,k}$, denoted specifically as $p_{t,k} = p_{t,k-1} + \delta p$.

To enhance the location accuracy of the method, we can perform multiple phase carrier phase measurements at the same observation point and use the weighted least squares method described in reference [35] to further reduce the impact of measurement errors on the effectiveness of the method. If $\sigma_{ji}$ represents the standard deviation obtained from multiple measurements of $u_{ji}$, the corresponding weight assigned to $u_{ji}$ is as follows.

$$w_{j,i} = \frac{1}{\sigma_{j,i}} \tag{20}$$

Add weights to Equation (12), which can be rewritten as:

$$WH\delta p = Wb \tag{21}$$

in which $W$ is given as follows.

$$W = \begin{bmatrix} w_{1,1} & & & \\ & w_{1,2} & & \\ & & \cdots & \\ & & & w_{M,6} \end{bmatrix} \tag{22}$$

For Equation (20), we directly apply the least squares solution Equation (18) of Equation (11) to obtain the weighted least squares solution of Equation (20) as follows.

$$\delta p = \left( H^T W^T W H \right)^{-1} H^T W^T W b \tag{23}$$

In this paper, the spoofing interference location is solved iteratively using Equation (23).

### 3.3. The CRB of the Method Proposed in This Article

In this section, we present the CRB for our proposed method, taking into account the presence of measurement errors in carrier phase measurements and array antenna positioning. CRB is a widely recognized tool that establishes a lower bound on the accuracy of unbiased estimators. According to Equation (8), the error in spoofing location accuracy is influenced by two components: the measurement error of the carrier phase for each array element and the positioning error of the antenna array itself. To derive the CRB for the proposed method, we introduce an observation vector $\hat{u} = [\hat{u}_{11}, \hat{u}_{12}, \cdots, \hat{u}_{ji}, \cdots, \hat{u}_{M6}]^T$, representing the carrier phase measurements. Here, $\hat{u}_{ji}$ corresponds to $u_{ji}$ in Equation (8) and encompasses the observation error. We define vector $\hat{u}$'s true value as $u = [u_{11}, u_{12}, \cdots, u_{ji}, \cdots, u_{M6}]^T$ and its measurement error vector as $\Delta u = \hat{u} - u$, where $\Delta u$ follows a zero-mean Gaussian distribution with a covariance matrix $Q_u = E[\Delta u \Delta u^T]$.

Similarly, we define the observation vector $\hat{v} = \left[ \hat{v}_{11}^T, \hat{v}_{12}^T, \cdots, \hat{v}_{ji}^T, \cdots, \hat{v}_{M6}^T \right]^T$ related to the position of the antenna array, where $\hat{v}_{ji}$ corresponds to $v_{ji}$ in Equation (8) and contains the observation error. We define vector $\hat{v}$'s true value as $v = \left[ v_{11}^T, v_{12}^T, \cdots, v_{ji}^T, \cdots, v_{M6}^T \right]^T$ and its measurement error vector as $\Delta v = \hat{v} - v$, where $\Delta v$ follows a zero-mean Gaussian distribution with a covariance matrix $Q_v = E[\Delta v \Delta v^T]$. According to the method in reference [36], we define vector $\Phi = [p_t^T, v^T]$, where $p_t$ represents the exact location of the target spoofing interference source. Given that $\Delta u$ and $\Delta v$ are independent of each other and follow zero-mean Gaussian distributions, the logarithm of the joint density function of $\hat{u}$ and $\hat{v}$, parametrized on $\Phi$, can be expressed as follows:

$$\begin{aligned} lnf(\hat{u}, \hat{v}, \Phi) &= lnf(\hat{u}, \Phi) + lnf(\hat{v}, \Phi) \\ &= K - \tfrac{1}{2}[\hat{u} - u]^T Q_u^{-1}[\hat{u} - u] - \tfrac{1}{2}[\hat{v} - v]^T Q_v^{-1}[\hat{v} - v] \end{aligned} \tag{24}$$

in which $f(\cdot)$ is the probability density function and the parameter $K$ associated with $Q_u$ and $Q_v$ is calculated as follows.

$$K = -\frac{1}{2}ln\left[ (2\pi)^{6M}|Q_u| \right] - \frac{1}{2}ln\left[ (2\pi)^{6M}|Q_v| \right] \tag{25}$$

According to the definition of CRB, the CRB equation for $\Phi$ is as follows.

$$CRB(\Phi) = -E\left[ \left( \frac{\partial^2 lnf(\hat{u}, \hat{v}, \Phi)}{\partial \Phi \partial \Phi^T} \right) \right]^{-1}_{(6M+3)\times(6M+3)} \tag{26}$$

It can be seen from Equation (25) that $CRB(\Phi)$ is a $6M + 3$ square matrix, in which only the upper left $3 \times 3$ submatrix corresponding to the CRB is of interest to us. For the convenience of expression, we convert Equation (26) into submatrix form shown as:

$$CRB(\mathbf{\Phi}) = \begin{bmatrix} R_1 & R_2 \\ R_2{}^T & R_3 \end{bmatrix}^{-1} \tag{27}$$

where $R_1$, $R_2$, and $R_3$ are defined as follows.

$$R_1 = E\left[\left(\frac{\partial^2 lnf(\hat{u}, \hat{v}, \mathbf{\Phi})}{\partial p_t \partial p_t{}^T}\right)\right] = \left(\frac{\partial u}{\partial p_t{}^T}\right)^T Q_u{}^{-1} \frac{\partial u}{\partial p_t{}^T} \tag{28}$$

$$R_2 = E\left[\left(\frac{\partial^2 lnf(\hat{u}, \hat{v}, \mathbf{\Phi})}{\partial p_t \partial v^T}\right)\right] = \left(\frac{\partial u}{\partial p_t{}^T}\right)^T Q_u{}^{-1} \frac{\partial u}{\partial v^T} \tag{29}$$

$$R_3 = E\left[\left(\frac{\partial^2 lnf(\hat{u}, \hat{v}, \mathbf{\Phi})}{\partial v \partial v^T}\right)\right] = \left(\frac{\partial u}{\partial v^T}\right)^T Q_u{}^{-1} \frac{\partial u}{\partial v^T} + Q_v{}^{-1} \tag{30}$$

Based on the inverse formula of the block matrix in [37], the following CRB calculation equation can be given.

$$CRB(p_t) = \left(R_1 - R_2 R_3{}^{-1} R_2{}^T\right)^{-1} = R_1{}^{-1} + R_1{}^{-1} R_2 \left(R_3 - R_2{}^T R_1{}^{-1} R_2\right)^{-1} R_2{}^T R_1{}^{-1} \tag{31}$$

*3.4. The Method Processing Flow*

Figure 7 illustrates the data processing flow description of the GNSS spoofing interference source localization method based on a moving array antenna. The steps of the method are as follows:

1.  We conduct real-time monitoring of the satellite identification code tracked by each tracking channel and determine whether there are two different channels tracking the same satellite. If the above conditions exist, it is judged spoofing interference is detected, and output the corresponding tracking channel vector $\mathbf{Cha} = [Cha1, Cha1', \cdots Chaj, Chaj', \cdots, ChaK, ChaK']$, where $Chaj$ and $Chaj'$ are channel numbers tracking the same satellite.

2.  Take any pair of tracking channels $Chaj$ and $Chaj'$ that track the same satellite signal, take any antenna array element $i$, and calculate the carrier phase double-difference between the other channels in $\mathbf{Cha}$ and the selected channel to obtain the carrier phase double-difference observation vectors $\Delta\mathbf{\Phi}_i^{Chaj} = [\Delta\varphi_i^{Cha1',Chaj}, \cdots \Delta\varphi_i^{Chaj'-1,Chaj}, \cdots, \Delta\varphi_i^{ChaK',Chaj}]$ and $\Delta\mathbf{\Phi}_i^{Chaj'} = [\Delta\varphi_i^{Cha1,Chaj'}, \cdots \Delta\varphi_i^{Chaj-1,Chaj'}, \cdots, \Delta\varphi_i^{ChaK,Chaj'}]$, where the expression of double-differenced carrier phase $\Delta\varphi_i^{Chaj'-1,Chaj}$ is the same as that of Equation (5).

3.  After obtaining the carrier phase double-difference, use the method in Section 3.1 to extract and confirm the spoofing interference signals and output the tracking channel vector $\mathbf{Cha}' = [Cha1', \cdots Chaj' \cdots, ChaK']$ corresponding to the deception satellite signals.

4.  Compare the carrier-to-noise ratio (C/N0) of the tracking signals of each channel in vector $\mathbf{Cha}'$, and take the deception satellite signal $Chamax'$ with the highest C/N0 among them.

5.  Collect carrier phase single-difference data of the deception signal $Chamax'$ along the spoofing location path and obtain the observation vector $\boldsymbol{\varphi}^{Chamax'} = [\varphi_{1,1}{}^{Chamax'}, \varphi_{1,2}{}^{Chamax'}, \cdots, \varphi_{j,i}{}^{Chamax'}, \cdots, \varphi_{M,6}{}^{Chamax'}]^T$, where the expression of carrier phase single-difference $\varphi_{j,i}{}^{Chamax'}$ is the same as that of Equation (6).

6.  After acquiring the carrier phase single difference data, employ the methodology elucidated in Section 3.2 to directly ascertain the precise location of the spoofing interference and yield the spatial coordinates of its source $p_t$.

We would like to give the following explanation for the practical application of the method:

- The double-differenced carrier phase-based spoofing interference extraction method enables an efficient separation between the real satellite signals and the spoofing signals, thereby facilitating the use of authentic satellite signals for determining the position of the array antenna itself.
- After implementing spoofing interference extraction, in principle, any satellite signal of the spoofing interference can be utilized to construct equation system (9). With increased signal strength, the accuracy of carrier phase measurement also improves, thus enhancing the effectiveness of the method; we select the satellite signal with the highest C/N0 for establishing the equation.
- The method presented in this paper is also applicable to multiple spoofing interference source localization scenarios; however, the applicable scenario requires that the located interfering source emits two or more spurious satellite signals. For multiple spoofing interference sources, the method in Section 3.1 of this paper can be used multiple times to extract multiple spoofing source signals. After the multiple spoofing sources are identified, the method described in Section 3.2 can be applied to each spoofing source to achieve efficient localization of multiple spoofing sources.

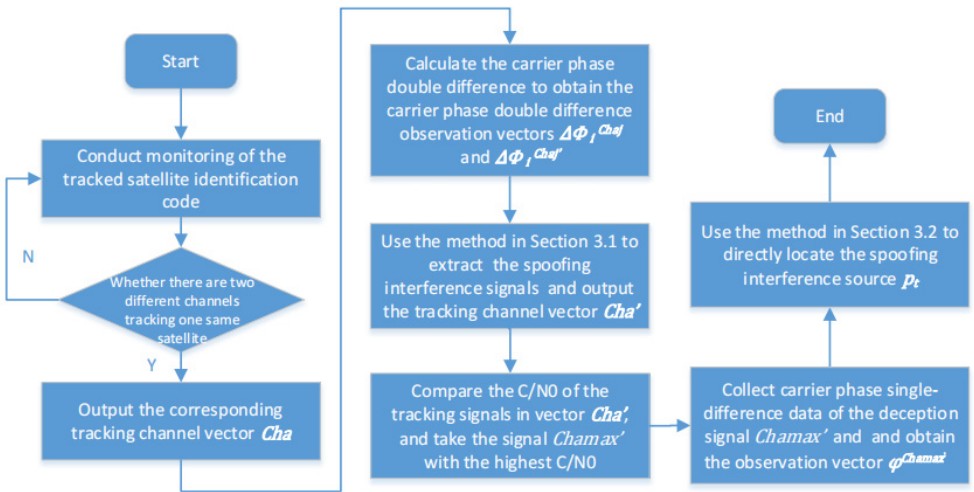

**Figure 7.** Flow chart of the proposed method.

## 4. Simulations and Results

The effectiveness of the proposed method in this paper is evaluated through simulations, which are performed using the MATLAB R2021b software platform. The effectiveness of the proposed spoofing interference extraction and confirmation method is examined in Experiment 1. The effectiveness of the suggested method for locating deception interference is assessed in Experiments 2 through 4. Experiment 2 compares and tests the localization effects of spoofing interference using the proposed approach and the two-step methods under different carrier phase measurement errors along five typical monitoring paths. Experiment 3 compares and tests the spoofing interference localization effects of our method and the better two-step method under different antenna positioning errors as well as varying carrier phase measurement errors. The spoofing interference localization effect is examined in Experiment 4, considering a varying number of observation points. The aforementioned simulation experiments examine the key factors that influence the effectiveness of the proposed method, and the results substantiate the effectiveness of the proposed method while providing guidance for its practical application.

### 4.1. Simulation Scenario

In this section, we describe a simulation scenario where four real satellites and one spoofing jamming source are set up. The coordinates of the real satellites are set to $[0\ 0\ 20,200,000]$, $[4,430,168\ 18,898,507\ 20,200,000]$, $[7,920,758 − 1,265,592\ 20,200,000]$, and $[−15,905,180\ 5,155,887\ 20,000,000]$, respectively, with the coordinate units being

meters. The subsequent coordinates in the article are also in meters. The simulation experiments employ the GPS L1 C/A signal (with a carrier frequency of 1575.42 MHz), while the four satellite transmission signals are denoted as Sta1–Sta4, respectively. The satellite signals emitted by the spoofing interference source are consistent with the real signals. The receiving antenna array adopts the 6 + 1 array antenna model in Section 2.1, and the spacing between array elements, denoted as $R$, is set to $\lambda/2$, where $\lambda$ represents the wavelength corresponding to the carrier frequency.

The schematic diagram of the simulation scenario is presented in Figure 8, with the interference monitoring area set to 1 km × 1 km × 200 m. To evaluate the method's location effectiveness under various monitoring paths, five typical array antenna-based signal monitoring paths are set in combination with practical application.

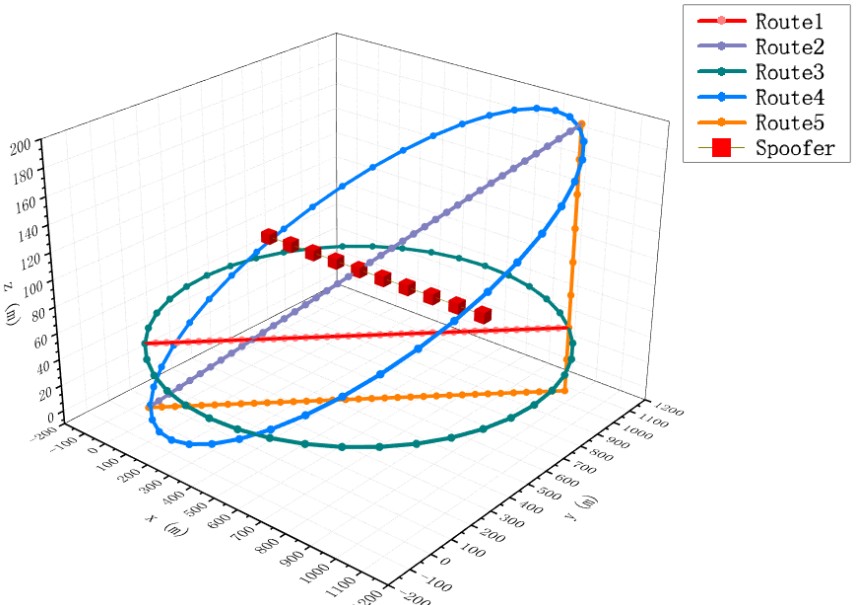

**Figure 8.** Schematic diagram of the simulation scenario.

Path 1 moves in a straight line from the initial coordinate [0 0 50], along the same height, to the final coordinate [1000 1000 50], with no relative displacement on the *z*-axis of the coordinate system. Path 2 proceeds in a straight trajectory from the initial coordinate [0 0 0], following the diagonal direction within the monitoring region, to the termination coordinate [1000 1000 200]. Path 3 is a circular trajectory of radius $500\sqrt{2}$, centered at [500 500 50], which exhibits no displacement along the *z*-axis. Path 4 is an elliptical trajectory that covers the entire surveillance area. The center of the ellipse is [500 500 100], and the normal of the plane of the ellipse is perpendicular to Path 2. Path 5 moves in a straight line from coordinate [0 0 0] to coordinate [1000 1000 0] and then from coordinate [1000 1000 0] to coordinate [1000 1000 200]. The five paths are each divided into 41 evenly spaced observation points to collect data, resulting in a total of 205 observation points.

In practice, we find that the monitoring path has a larger impact on the signal location and search effect, but the relevant study content is smaller. Thus, typical monitoring paths are established in this paper to test the performance of the algorithm under different paths and, thus, guide the practical application of the method. The set of five paths are common paths in engineering practice that have relatively acceptable monitoring performance under the conditions of a given monitoring area. The characteristics of each path are described as follows:

- Path 1 corresponds to the setting of covering the monitoring area with the shortest path at a fixed monitoring height under the condition that the monitoring platform

height cannot be raised (such as limited airspace height or ground monitoring equipment scenarios).

- Path 2 corresponds to the setting of covering the monitoring area with the shortest path under the condition that the height of the monitoring platform can rise.
- Path 3 corresponds to the setting of covering the monitoring area with the maximum circular area under the condition that the monitoring platform height cannot be raised.
- Path 4 corresponds to the setting of covering the monitoring area with the maximum elliptical area under the condition that the height of the monitoring platform can rise.
- Path 5 corresponds to only supplementing the displacement setting in the z-direction on the basis of Path 1 under the condition that the height of the monitoring platform can rise.

To comprehensively evaluate the performance of the method, we localize the interference sources at different locations. The red squares in Figure 8 show the locations of spoofing interference sources, where the interference path is set to be a straight path. The starting and ending points are [100 500 100] and [100 500 1000], respectively. Equidistant sampling is conducted between them, and a total of 10 points are taken as interference source localization test points.

*4.2. Experiment 1: Spoofing Extraction Effects Based on the Double-Difference Carrier Phase*

Considering that spoofing interference extraction has no restriction on the observation paths, this experiment uses 205 observation points from all five paths to test the effectiveness of the deception interference extraction method proposed in this paper. The location of the spoofing interference source is set to [500 500 100], which is the center position of the interference monitoring region. The mean value of the carrier phase measurement error is set to 0, and the standard deviation is set to $0.005 \times 2\pi$ rad.

Prior to simulation testing, distinguishing between spoofing interference signals and genuine satellite signals is unattainable. The spoofing interference extraction methodology in this paper necessitates the concurrent utilization of eight signal tracking channels to monitor both the authentic satellite signal and the spoofed interference signal, thus enabling carrier phase measurements on all eight satellite signals. We numbered the 8 tracking channels as 1–8, with channels 1 and 2 receiving the Sta1 signal, channels 3 and 4 receiving the Sta2 signal, channels 5 and 6 receiving the Sta3 signal, and channels 7 and 8 receiving the Sta4 signal.

The carrier phase measurements between antenna elements 1 and 0 are utilized for the carrier phase single-difference calculation in the experiment. Subsequently, the single-difference measurement values of signal channels 1 and 2 are subtracted from those of signal channels 3–8, respectively, to compute the double-differenced carrier phase.

The comparison between the double-differenced carrier phase observation values and the set threshold is depicted in Figure 9. Figure 9a presents the double-differenced carrier phase results for channels 3–8 and channel 1, while Figure 9b displays the results for channels 3–8 and channel 2. The horizontal axis in Figure 9 represents the sequential numbering of the observed points, while the vertical axis corresponds to the double-difference carrier phase measurements at these points, denoted in units of $2\pi$ rad. We set the double-difference carrier phase threshold *Th* for spoofing interference extraction as $0.04 \times 2\pi$ rad.

Although there is a situation in Figure 9a where sporadic double-difference measurement results are below the threshold, it lacks continuity. In Section 3.1, the method stipulates that there must be five consecutive observation points with double-difference carrier phase measurements smaller than the set threshold in order to determine that the two satellite signals are from the same spoofing interference source, thus avoiding such false alarms. The double-difference measurement values of 4–2, 6–2, and 7–2 in Figure 9b fall below the threshold, indicating that the satellite signals tracked by channels 2, 4, 6, and 7 originate from the same deceptive interference signal. The effectiveness of the spoofing interference extraction method in this paper is tested using 205 observations from the five

paths. Every five observation points is a test, making a total of 41 tests. The results are statistically analyzed, and the probability of successful extraction of spoofing interference signals is 100%, and the missing alarm rate is 0%. The experimental results demonstrate that the extraction of spoofing interference based on the method double-differenced carrier phase can effectively separate the spoofing signals from the authentic signals.

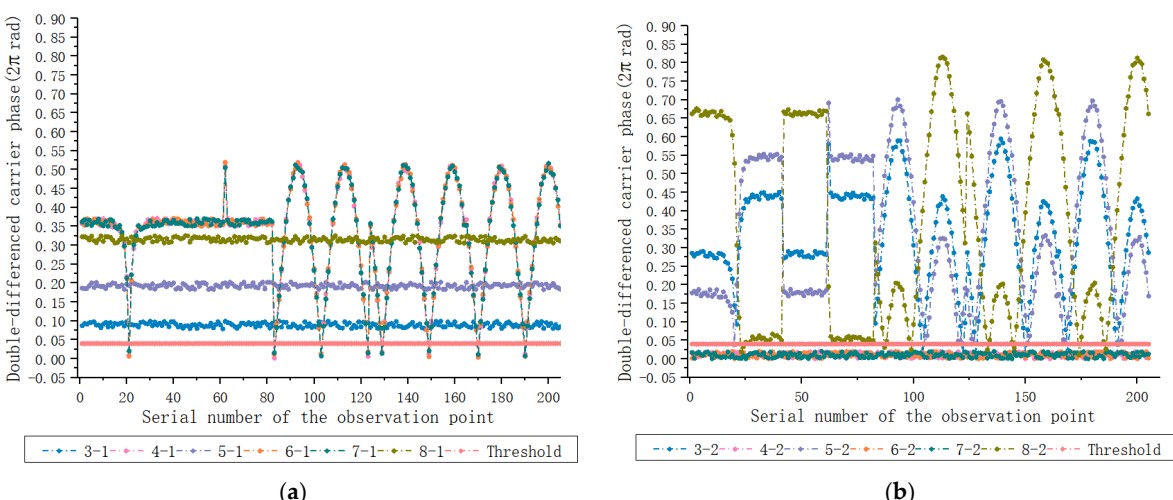

**Figure 9.** Comparison of the double-difference carrier phase observables with the set threshold. (**a**) Double-difference results with channel 1; (**b**) Double-difference results with channel 2.

It is worth mentioning here that multi-path can affect the effectiveness of the method in this paper and generate false positives in real-world applications, and we would like to reduce the effect of multi-path on the effectiveness of the method by improving the application details in this paper. Multi-path may affect the effectiveness of the proposed method in a certain multi-path region for a short period of time, but as the monitoring platform moves over a large area, it can avoid or reduce the effect of multi-path on the effectiveness of the proposed method. In order to achieve effective deception, the spoofing interference source needs to transmit deception signals for a long time so it can appropriately increase the number requirement of observation points that meet the above conditions (for example, the double-differenced carrier phase observations of 10 consecutive observation points are less than the set threshold), or meet the continuous observation distance requirement while meeting the continuous observation points requirement (for example, the double-differenced carrier phase observations of 100 m consecutive observation distance are less than the set threshold), enabling the method to achieve deception interference confirmation over a large observation range, so as to eliminate or weaken the influence of multi-path on the effectiveness of the proposed method to a certain extent.

### 4.3. Experiment 2: Spoofing Localization Effects at Different Carrier Phase Measurement Errors

This experiment tests the effectiveness of different GNSS spoofing interference source localization methods under different carrier phase measurement errors along the above five paths. The experimental comparison methods include the method proposed in this article, the global search method for the cost function of Equation (16), and two typical two-step methods. In the two-step comparison methods, the measurements of the arrival angle of deceptive interference signal by array antennas are both implemented using the approach described in reference [38]. Two-step Method 1 employs the Beamsum Criterion principle from reference [39] to locate the spoofing interference source, while Two-step Method 2 utilizes the least squares principle from reference [39] for locating the deceptive source.

To comprehensively evaluate the performance of the proposed method, Experiment 2 tests the effects of the proposed method under different spoofing interference source location settings and performs statistics on the location results. The spoofing source locations

setup are shown in Figure 8. According to references [40,41], while ensuring effective tracking, the setting range for the standard deviation of carrier phase measurements in this experiment is $0.005 \times 2\pi$ rad $- 20 \times 0.005 \times 2\pi$ rad, with a mean value set at 0 and measured in radians. The standard deviation of the self-positioning error of the array antennas is set to 1 with a mean value of 0 and units are meters.

The second experiment involves conducting 100 tests for each carrier measurement error setting, wherein all 10 interference sampling points in the interference path shown in Figure 8 are located for each test. Subsequently, the root mean square errors (RMSEs) of the aforementioned 1000 location results are calculated. In each test, we conducted 10 carrier phase single-difference observations and constructed matrix $W$ using Equations (20) and (22). The experimental results are presented in Figure 10, where Figure 10a–e illustrates the comparative outcomes among different methodologies across five paths. The horizontal axis of the figure represents the standard deviation of the carrier phase measurement error, measured in $0.005\pi$ rad, while the vertical axis represents the localization RMSE, measured in meters. Simulation results demonstrate that as the carrier phase measurement error increases, the proposed method exhibits significantly superior location accuracy compared to the two-step methods, and it closely approximates the CRB and the global search method. This demonstrates that the proposed method is more adaptable to carrier measurement errors and is more suitable for spoofing interference monitoring applications in the dynamic scenario of this paper. Figure 10f compares the location performance of our method under five monitoring paths, among which straight paths 1, 2, and 5 are better than circular paths 3 and 4. Path 5 has the best performance among the straight paths due to its displacement in both horizontal and vertical directions. Therefore, we choose path 5 as the recommended path for our method.

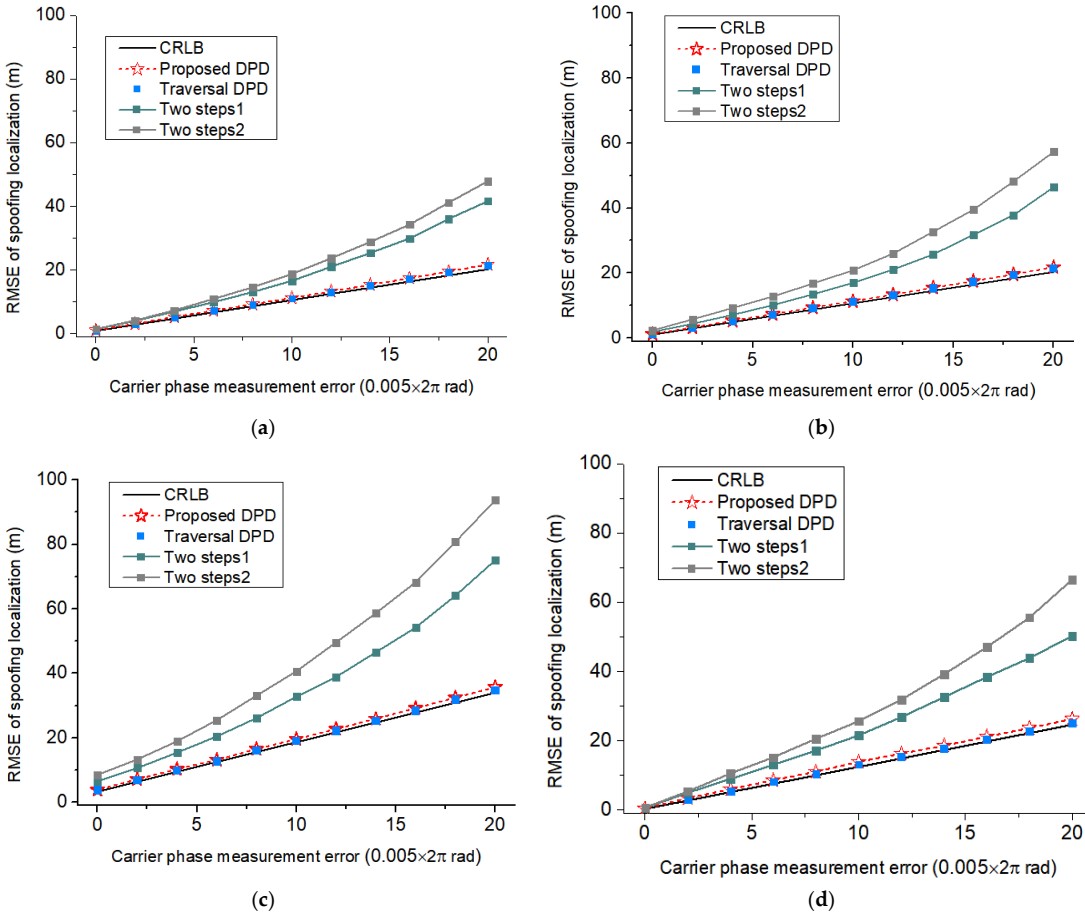

**Figure 10.** *Cont.*

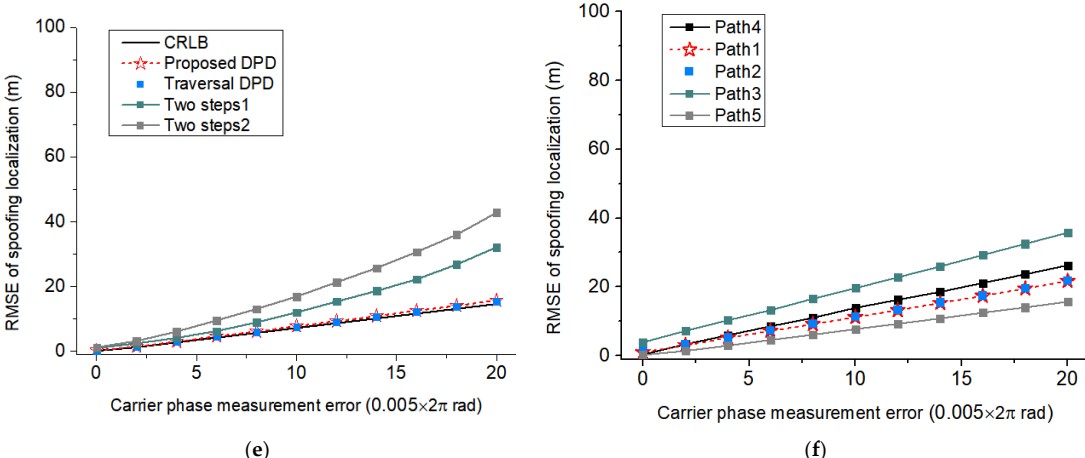

**Figure 10.** Test results from Experiment 2. (**a**) Comparison of localization effects of various methods under Path 1; (**b**) Comparison of localization effects of various methods under Path 2; (**c**) Comparison of localization effects of various methods under Path 3; (**d**) Comparison of localization effects of various methods under Path 4; (**e**) Comparison of localization effects of various methods under Path 5; (**f**) Comparison of our method under five monitoring paths.

In this experiment, we recorded the number of iterations of the method and the normalized value of the cost function Equation (16) while the method was running and statistically analyzed the mean value of the normalized cost function for 1000 experiments. Figure 11 shows the relationship between the number of iterations and the mean value of the normalized cost function under the five test paths of this experiment. The results show that the five test paths can converge after six iterations. To ensure the effectiveness of the algorithm, the number of iterations in this paper is set to eight.

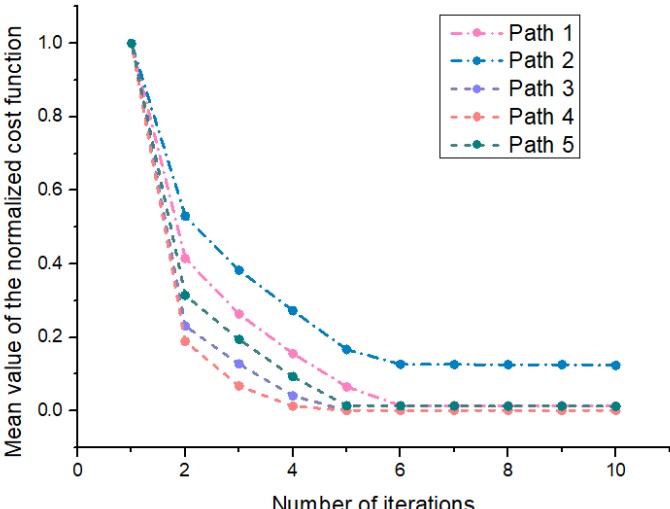

**Figure 11.** Statistical results for the normalized cost function under different number of iterations.

*4.4. Experiment 3: Spoofing Localization Effects under Different Antenna Positioning Errors*

This experiment tests the spoofing interference localization effects of our method under different antenna localization errors and carrier phase measurement errors and selects the Two-step Method 1 with better localization effect as the comparison method. The spoofing interference source setup, carrier phase measurement error setup, and localization accuracy statistics for the spoofing interference source in the experiment are consistent with those in Section 4.2.

The mean values of the antenna positioning errors in Experiment 3 are all set to 0, and the standard deviation values are set to 1, 5, 15, 20, 30, and 50, respectively, in meters. Here, the standard deviation settings of 1 and 5 correspond to the positioning accuracy that can be achieved by a typical 7-element null antenna under static and low-speed motion, the settings of 15 and 20 correspond to the positioning accuracy of the array antenna under conventional motion, and the settings of 30 and 50 correspond to the positioning accuracy of the array antenna under high-speed motion.

The experimental results are shown in Figure 12, where Figure 12a,b shows the spoofing interference localization effects of the proposed method and the Two-step Method 1 under different antenna positioning error settings, respectively. The horizontal and vertical coordinate settings in Figure 12 are consistent with those in Figure 10. The simulation results show that the proposed method has stronger adaptability to antenna positioning errors compared to the two-step method, making it more suitable for deception interference monitoring applications in dynamic scenarios in this paper. As the antenna array positioning error increases, the main factor affecting the spoofing location effect of the method in this paper changes from carrier phase measurement error to antenna positioning error.

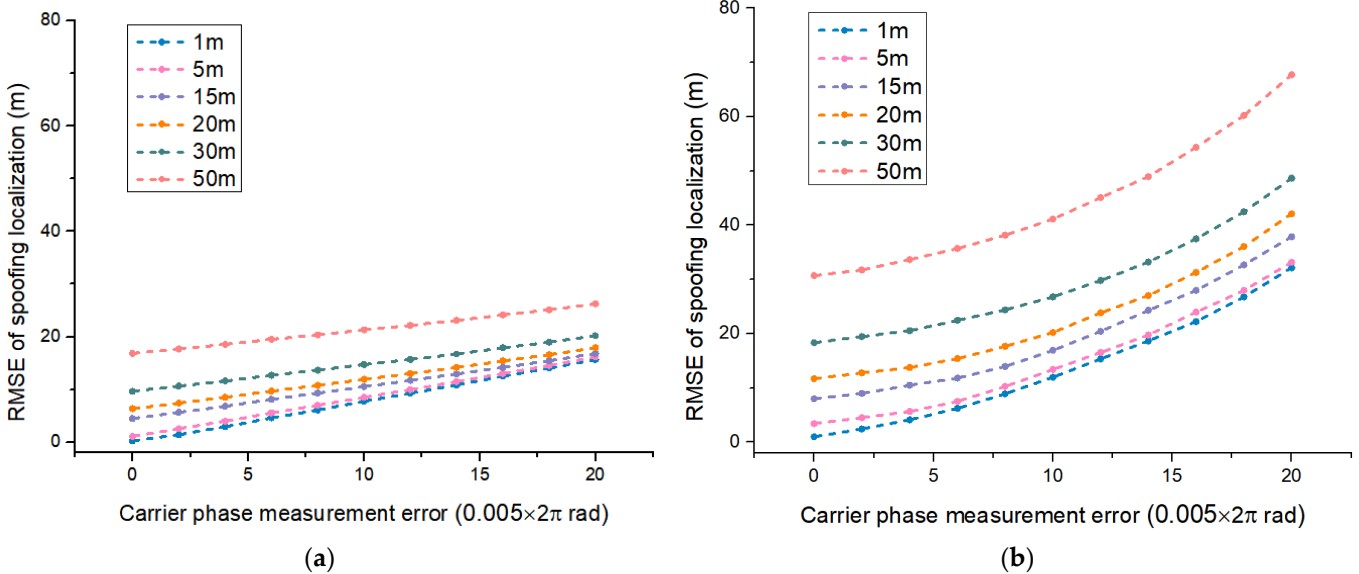

**Figure 12.** Test results from Experiment 3. (**a**) Comparison of localization effects under different antenna positioning error settings of the proposed method; (**b**) Comparison of localization effects under different antenna positioning error settings of the Two-step Method 1.

### 4.5. Experiment 4: Spoofing Localization Effects Considering Varying Numbers of Observation Points

Experiment 4 tests the effectiveness of the proposed method in spoofing interference localization under different numbers of observation points. The spoofing interference source settings, carrier phase measurement error settings, antenna positioning error settings, and the positioning accuracy statistical methods for deception interference sources in the experiment are consistent with Section 4.2. The experimental path used in the experiment is recommended Path 5, with the number of observation points set to 11, 21, 41, 61, and 81, respectively, and each observation point number setting is uniformly sampled of Path 5. The experimental results are shown in Figure 13, where the horizontal and vertical coordinate settings are consistent with those of Figure 10.

Simulation results show that the spoofing interference localization effect of the proposed method improves as the number of observation points increases. However, as the number of observation points increases, the improvement of the spoofing interference localization effect decreases, and the increase in the number of observation points leads to a significant increase in computational complexity. Considering the cost of the implementa-

tion of the method, the recommended number of observation points for the monitoring region set in the experimental scenario in this paper is 21 or 41.

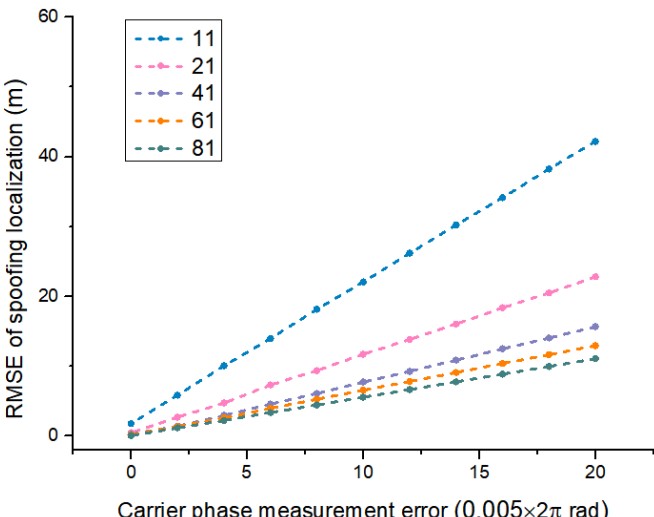

**Figure 13.** Comparison of localization effects at varying numbers of observation points.

### 4.6. Experiment 5: Spoofing Localization Effects Considering Compatibility at Different Frequency Points

The GNSS navigation band is not a continuous broadband but consists of multiple narrow bands with fixed carrier frequencies, so we typically use a 6 + 1 model to cover a narrow GNSS band and a dual 7-element design to cover a dual GNSS band in practice. For example, an antenna array commonly used in our practical applications that covers the navigation frequency bands of GPS L1 (1575.42 MHz), BDS B1 (1561.098 MHz), and BDS B3 (1268.52) generally consists of two circles of array elements. The antenna elements in the outer circle are arranged with the 1/2 wavelength of the BDS B3 frequency point as the baseline, while the antenna elements in the inner circle need to consider both GPS L1 and BDS B1. To avoid the occurrence of integer ambiguity, we choose half of the shorter wavelength GPS L1 wavelength as the baseline for layout, so we should consider the compatibility of the above layout method at the BDS B1 frequency point.

Experiment 5 tests the effectiveness of the proposed method for the localization effect of deception interference sources at GPS L1 frequency point and BDS B1 frequency point under the above antenna setting. The spoofing interference source settings, carrier phase measurement error settings, antenna positioning error settings, and the positioning accuracy statistical methods for deception interference sources in the experiment are consistent with Section 4.5. The numbers of observation points are set to 21 and 41, respectively. The experimental results are shown in Figure 14, where the horizontal and vertical coordinate settings are consistent with those of Figure 10.

Simulation results show that under the proposed GPS L1 antenna design, the localization effect of the proposed method is essentially the same for both BDS B1 and GPS L1 frequency point spoofing interference sources, while the BDS B1 effect is slightly reduced. The proposed method is compatible with GPS L1 and BDS B1 frequency points, which enables efficient localization of spoofing interference at the aforementioned frequency points.

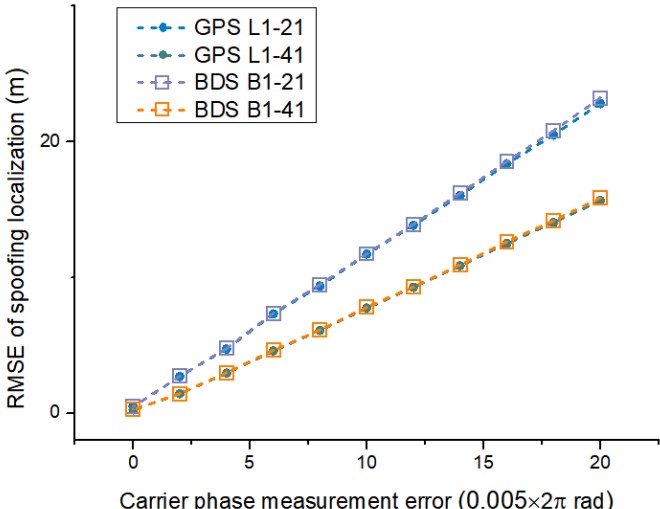

**Figure 14.** Comparison of localization effects at different frequency points.

## 5. Conclusions

In this paper, we propose a GNSS spoofing interference source localization method based on a moving array antenna. The method has strong usability and is suitable for navigation nulling antenna hardware platforms, enabling it to have both anti-interference and deceptive interference detection and localization capabilities. This paper elaborates on the antenna model and navigation signal acquisition and tracking methods used in the implementation of the proposed method. Based on this, GNSS spoofing interference source localization is proposed in this paper. The proposed method first uses a spoofing interference extraction method based on a double-differenced carrier phase to effectively separate the deceptive and real signals. After the spoofing interference is extracted, the original carrier phase single-difference data of the spoofing signal from multiple observation points is fused through a moving antenna array to directly localize the spoofing interference. The procedure for calculating the CRB of the proposed method is also provided.

Simulations are used to test the effectiveness of this approach. Simulation results show that this approach effectively solves the spoofing interference extraction and location problems. Simulation results also show that the proposed method is nicely adapted to carrier phase measurement errors and antenna positioning errors and that the proposed method is close to CRB. Based on the test results, the recommended monitoring path and the number of observation points for the proposed method are given. Subsequent work will test the method in combination with a physical platform to test the detection and location effects of spoofing interference in real-world scenarios. At the same time, the effectiveness of the method in the multi-path scenario will be the focus of our physical platform tests. The effectiveness of multi-path weakening means, and the effects of the parameter settings (such as the number requirement of observation points) will be explained in combination with the physical platform test effect of this method.

**Author Contributions:** Conceptualization, Z.Y. and G.L.; Methodology, R.L. and Q.C.; Resources, Q.Z. All authors have read and agreed to the published version of the manuscript.

**Funding:** This work was supported by the National Key Research and Development Program of China under Project 2018YFB0505100.

**Data Availability Statement:** Data are contained within the article.

**Conflicts of Interest:** The authors declare no conflict of interest.

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
