# Peer review of "Localization of GNSS Spoofing Interference Source Based on a Moving Array Antenna"

_remotesensing, doi:10.3390/rs15235497_

Round 1

Reviewer 1 Report

Comments and Suggestions for Authors

Please see the attached comments.

Comments on the Quality of English Language

The quality of English languag can be further improved, and proof-reading is suggested. 

Author Response

It's great to hear from you! I have been highly inspired by the comments you have made. We have thoroughly addressed each of your comments and have carefully revised the manuscript in light of these invaluable suggestions. Please refer to the attachment for the response letter. Thank you so much once again for your extremely helpful suggestions.

Reviewer 2 Report

Comments and Suggestions for Authors

The paper discusses locating GNSS spoofing interference sources using a moving array antenna. The novelty is the use of a double-differenced carrier phase to identify the location of the jammer/spoofer. An element-nulling antenna is proposed with the separation of elements based on the received signal frequency.

The method was verified only through simulation.

The paper has good background research, an extensive reference list and good English.

The paper is well written, yet given that only simulation data is presented I would suggest adding the following:

* fig 8 and 10 would benefit from the consistent axis scale

* It is not clear how efficient is proposed antenna setup with different frequencies, as the design is based on halfwave length. Explaining (or simulating) those results would increase the strength of the paper.

* The multipath effect (that can create both multi-directional DOA and non-line of sight) is not discussed, hence statement on page 14 (false alarm rate is 0 percent and false alarm rate is 0 percent) seems premature - MP is likely to cause a false alarm. Explaining the effect of multipath would strengthen the paper.

Author Response

Thank you so much for your positive comments and thoughtful suggestions! I have been highly inspired by the comments you have made. We have thoroughly addressed each of your comments and have carefully revised the manuscript in light of these invaluable suggestions. Please refer to the attachment for the response letter. Thank you so much once again for your extremely helpful suggestions.

Reviewer 3 Report

Comments and Suggestions for Authors

1. In several places in the manuscript the Figures are referred to as Figure, and somewhere to as Fig. (e. g. Figure 2 vs. Fig. 2). I would recommend a embrace a consistent approach: chose one option and purse it throughout the paper. 

2. To be geometrically consistent with the explanation given in rows 121 – 128, a would recommend to modify the Figure 2 in a way that the axis 'y' points to the middle between the elements 2 and 3, and that the angle γ2 points to one specific element (e.g. element 2).

3. There is no 'Section 2.1.1.', please correct to 'Section 2.1) (row 130).

4. I would prefer so see the full names of abbreviations (RF, IF, AD) given in rows 135 and 136.

5. Explanation about the quantity ? is missing.

6. Pay attention to brackets in equation (4) in row 221.

7. Align the orientation of the axis 'y' in Figure 5 with those in Figure 2.

8. It seams that the dimension in the equations (6) and (7) are not correct: on the LHS we have [m], on the RHS there is [m2s-1]. Please check the equations (6) and (7)!

9. What is the meaning of matrix G in the equation (17)? Please, check!

10. I would change the title of the subsection 3.4 from 'The Method Flow of This Paper' to just 'The Method Processing Flow'.

11. In the row 493 the usage of '%' symbol is recommended (100 percent vs. 100 %).

12. Row 519: change from 'is calculated' to 'are calculated'

Comments on the Quality of English Language

Some minor corrections have to be done (as indicated above).

Author Response

(The authors gave the same response as above.)

Round 2

Reviewer 3 Report

Comments and Suggestions for Authors

The manuscript has been improved, the overall quality has been brought to a higher level.